# A Sub-Clone of RAW264.7-Cells Form Osteoclast-Like Cells Capable of Bone Resorption Faster than Parental RAW264.7 through Increased De Novo Expression and Nuclear Translocation of NFATc1

**DOI:** 10.3390/ijms21020538

**Published:** 2020-01-14

**Authors:** Laia Mira-Pascual, Anh N. Tran, Göran Andersson, Tuomas Näreoja, Pernilla Lång

**Affiliations:** 1Department of Laboratory Medicine, Division of Pathology, Karolinska Institutet, Alfred Nobels Allé, 8, SE-141 52 Stockholm, Sweden; laia.mira.pascual@ki.se (L.M.-P.); anh.tran.1@ki.se (A.N.T.); goran.andersson@ki.se (G.A.); 2Musculoskeletal Group, Institute of Medical Sciences, University of Aberdeen, Aberdeen AB 252ZD, UK

**Keywords:** Osteoclast, RAW264.7, tartrate-resistant acid phosphatase, osteoclastogenesis, bone resorption, NFATc1, differentiation

## Abstract

The murine macrophage cell line RAW264.7 is extensively used as a progenitor to study osteoclast (OC) differentiation. RAW264.7 is a heterogeneous cell line, containing sub-clones with different abilities to form OCs. The aim of this study was to identify characteristics within the heterogeneous RAW264.7 cells that define sub-clones with an augmented ability to form bone-resorbing OCs (H9), as well as sub-clones representing non-OCs (J8). RAW264.7 sub-clones were isolated by single cell cloning. Selection was based on TRAP/cathepsin K expression in sub-clone cultures without added RANKL. Sub-clones before and after differentiation with RANKL were assayed for multiple OC-characteristics. Sub-clone H9 cells presented a higher expression of OC-markers in cultures without added RANKL compared to the parental RAW264.7. After 6 days of RANKL stimulation, sub-clone H9 cells had equal expression levels of OC-markers with RAW264.7 and formed OCs able to demineralize hydroxyapatite. However, sub-clone H9 cells displayed rapid differentiation of OC already at Day 2 compared to Day 4 from parental RAW264.7, and when cultured on plastic and on bone they were more efficient in resorption. This rapid differentiation was likely due to high initial expression/nuclear translocation of OC master transcription factor, NFATc1. In contrast to H9, J8 cells expressed initially very low levels of OC-markers, and they did not respond to RANKL-stimulation by developing OC-characteristics/OC-marker expression. Hence, H9 is an additional clone suitable for experimental setup requiring rapid differentiation of large numbers of OCs.

## 1. Introduction

Osteoclasts (OCs) are responsible for degrading the bone matrix, thereby initiating bone repair, which allows osteoblasts to build new bone and thus maintain the integrity of the bone tissue [1,2]. An important tool for understanding OC biology is the use of in vitro models of OC differentiation and activation from OC progenitors/precursors. Currently, sources of OC precursors include primary cultures of mouse bone marrow macrophages, spleen macrophages or human CD14+ monocytes [3,4,5], as well as immortalized cell lines [6,7] such as the ER-Hoxb8 [8] and RAW264.7 cell lines [9].

Common for both in vivo and in vitro OC differentiation is their dependence on two cytokines, M-CSF and receptor activator of NF-κB ligand (RANKL) [10,11,12,13,14,15]. OC precursors being stimulated with RANKL in combination with M-CSF leads to TRAF6 activation and the downstream triggering of several pathways (e.g., NF-kB, MEK, MKK6, MKK7), resulting in the activation of the master OC transcription factor, NFATc1, which is also activated though RANKL-dependent Ca^2+^ signaling [16]. This activation starts a program in the OC progenitor with increased expression of OC markers, such as tartrate-resistant acid phosphatase (TRAP), cathepsin K (CtsK), β3 integrin and calcitonin receptor (CTR), ending with OC fusion and differentiation [17].

One commercially available OC precursor for generating in vitro OC-like cells is the mouse macrophage cell line RAW264.7 [9,18] which was originally derived from ascites of a tumor induced by the Abelson leukemia virus in a male BALB/14 mouse. RAW264.7 shows a stable [19] mature adherent macrophage phenotype that, in response to RANKL stimulation, forms multinucleated TRAP-positive OC-like cells. Although widely used, it is debated whether OC-like cells derived from RAW264.7 can resorb bone—that is, if they exhibit biologically relevant OC functions resembling primary cells [20]. RAW264.7s have been reported to both acidify the bone matrix as well as degrade collagen, but the combination present in bone is harder to perforate. Still, RAW264.7-derived OC-like cells have several advantages over primary macrophage-derived OCs, making them an interesting in vitro model to use as they are an immortalized cell-line with a simple culture protocol and can be used to generate an indefinite number of OC-like cells rapidly for high throughput assays, and also, they secrete M-CSF [13,14,15], thus abolishing the need to add exogenous M-CSF during OC differentiation. Furthermore, they are easier to transfect compared to primary OCs or OC-precursors [21] and they are capable of acidification to demineralize hydroxyapatite-coated surfaces [22]. Despite differences in gene as well as protein expression patterns, recent investigations have indicated that, after RANKL-stimulation, RAW264.7s do resemble OCs [22,23].

The RAW264.7 cell line is heterogeneous, as macrophages typically are, and this introduces a set of problems in reproducibility of data between laboratories, cell batches and even experiments [24]. However, it is hypothetically possible that OC-like precursors with a different capacity to form functional OCs are present in the RAW264.7 cell line. Thus, by screening RAW264.7-derived sub-clones it might be possible to isolate OC-precursor clones, which more resemble OC upon RANKL-stimulation, than the large mixture of clones in RAW264.7. One attempt to isolate different OC precursor populations present in RAW264.7 cells has been performed [22] using single cell cloning. In this study, RAW264.7-clones with different capacities to form TRAP-positive multinucleated cells were isolated and characterized with regard to gene expression patterns and functional characteristics.

The aim of this study was therefore to investigate whether more homogenous OC-precursor sub-clones with the capacity to acidify as well as degrade collagen could be selected from RAW264.7 using a simpler method. Here, we present a RAW264.7-derived sub-clone, H9, with the ability to form TRAP-positive multinucleated OC capable of demineralization and collagen degradation faster than RAW264.7 due to increased mRNA expression and nuclear translocation of NFATc1. Furthermore, we present a thorough characterization of the attributes that lead to efficient OC-differentiation. Moreover, H9 is an additional experimental model in applications requiring rapid differentiation of large amounts of OCs. 

## 2. Results

### 2.1. Selection of RAW264.7 Sub-Clones and Gene Expression of OC Markers, TRAP and Cathepsin K

This study was initiated to explore the heterogeneity of the RAW264.7 cell line by single cell cloning, with an aim to select a clone presenting OC-like characteristics and another clone not presenting OC-like properties. A total of 24 clones were isolated using single cell cloning from parental RAW264.7 cells. A simple gene expression screening for OC markers, TRAP and CtsK in unstimulated cells (i.e., in the absence of RANKL) was performed. The clones were ranked according to an increasing expression of TRAP (Figure 1A). Clones H9.2d3, H4.2 and H9.2d5 ranked highest and of these H9.2d5 (hereafter referred to as H9) was selected, because it was further propagated in the dilution series and therefore had higher clonal purity (Figure 1A). Clones with a lower TRAP and CtsK expression than the parental RAW264.7 were scarce and we identified only one such clone, J8.2g6 (Figure 1A). 

H9 represented a possible OC-precursor candidate and J8.2g6 (hereafter referred to as J8) was selected due to it having the least resemblance to an OC-precursor, since it was the only clone isolated with lower TRAP and CtsK gene expression compared to the parental RAW264.7. New cultures of H9 and J8 without RANKL confirmed by RT-qPCR that H9 had higher TRAP (~20 times) and CtsK (~60 times) gene expression compared to unstimulated parental RAW264.7, while unstimulated J8 had lower TRAP (~0.07 times) and CtsK (~0.3 times) gene expression of these markers (Figure 1B). After stimulation with RANKL for 4 days, both RAW264.7 and H9 showed elevated levels of TRAP gene expression to approximately the same degree (~400 times) compared to unstimulated parental RAW264.7, whereas the CtsK gene expression in RANKL-stimulated H9 was significantly higher (~3.7 times) than in RANKL-stimulated RAW264.7 (Figure 1B). Upon RANKL stimulation, J8 showed a slightly smaller increase in TRAP and a significantly smaller increase in CtsK mRNA compared with H9 and parental RAW264.7 (Figure 1B).

In response to RANKL stimulation, H9-clone formed multinucleated TRAP-positive OC-like cells with a similar frequency to parental RAW264.7. In contrast, J8 formed few TRAP-negative small multinucleated cells, and the cultures consisted predominantly of mononuclear cells (Figure 1C). In unstimulated H9-cultures, there were occasional TRAP-positive multinuclear cells, a minor group of TRAP-positive mononuclear cells and a few TRAP-negative multinuclear cells (Figure 1C). In unstimulated RAW264.7-cultures, there were a few TRAP-negative multinuclear cells, but no TRAP-positive cells. In unstimulated cultures of J8, multinuclear cells were fewer in number, but the cell density appeared to be higher compared with H9 and parental RAW264.7 (Figure 1C).

### 2.2. RANKL-Stimulated H9 and RAW264.7 Form Resorbing Osteoclast-Like Cells While J8 Does Not

Having established that H9, as well as RAW264.7, expressed late stage OC-markers in higher levels than J8, the sub-clones H9 and J8 along with parental RAW264.7 were investigated for OC functions (i.e., demineralization, inability to phagocytose, formation of sealing zones and resorption pits and capacity to degrade collagen).

Firstly, the ability of RANKL-stimulated RAW264.7, H9 and J8 clones to dissolve hydroxyapatite was investigated in an assay for acidification capacity of mineralized extracellular matrix. The results show that OCs derived from RAW264.7 and H9 had a similar capacity to acidify, while RANKL-stimulated J8 did not dissolve hydroxyapatite to a significant extent (Figure 2A,B).

Next, the ability to phagocytose, a macrophage characteristic not present in OCs, was investigated in cultures of RAW264.7, H9 and J8 +/− RANKL-stimulation (Figure 2C). Using a phagocytosis assay based on the uptake of fluorescent *E. coli* after 5 days of culture without RANKL-stimulus, RAW264.7, H9 and J8 all phagocytosed to the same extent (no significant difference; Figure 2C). However, after 5-day RANKL-stimulation, RAW264.7s and H9s had lost their ability to phagocytose, while RANKL-stimulated J8s were still able to phagocytose to the same extent as in the absence osteoclastogenic stimulus (Figure 2C).

Differentiated OCs form specific microdomains (e.g., sealing zones, ruffled borders), and on the bone substrate make resorption pits. Electron micrographs of the sealing zone and ruffled border in RAW264.7s are shown in Appendix A. Therefore, the ability of RAW264.7, H9 and J8 to form sealing zones and resorption pits was investigated. OC-like cells derived from RANKL-stimulated H9 had a higher number of sealing zones per mm^2^ compared to ones derived from RAW264.7 (Figure 2D) and RANKL-stimulated J8 formed 50% fewer sealing zones compared to RAW264.7 (Figure 2D and Appendix A). Using bisphosphonate staining to identify resorption pits revealed that OCs derived from RANKL-stimulated H9 and RAW264.7 displayed a similar percentage of resorption pit area at Day 6 (Appendix A), but H9 resorbed deeper pits (Appendix A) and thus more bone volume (Figure 2E,F). Furthermore, the majority of the areas resorbed by parental RAW264.7s were shallow and could be interpreted as merely as acidification of the surface, whereas the resorption pits formed by H9 were several micrometers deep. Conversely, RANKL-stimulated J8 did not form resorption pits at all (Figure 2E,F and Appendix A). Moreover, H9 had formed resorption pits already at Day 3, while RAW264.7s did not catch up until Day 6 (Appendix A). Moreover, analysis of collagen breakdown products in the form of CTX-I fragments showed that the cells did resorb the organic bone matrix and thus behaved like functional OCs, and also that resorption was significantly augmented in RANKL-stimulated H9 and RAW264.7 compared to unstimulated H9, RAW264.7 and J8, as well as RANKL-stimulated J8 (Figure 2G).

### 2.3. Gene Expression of Specific Osteoclast Markers Are Elevated in Unstimulated H9 Compared to RAW264.7

To understand what might cause the functional difference, we investigated the gene expression patterns of the sub-clones established OC-markers implicated in various events during osteoclastogenesis, fusion, adhesion and demineralization were assessed in RAW264.7, and the sub-clones H9 and J8 unstimulated and RANKL-stimulated conditions.

In unstimulated H9, the key OC fusion protein OC-STAMP displayed higher expression levels (~70 times) compared to parental RAW264.7. However, after 4 days of RANKL stimulation, the expression of OC-STAMP mRNA had doubled in H9 and the parental RAW264.7 reached similar expression levels as H9 (Figure 3A). Also, gene expression of another key fusion protein DC-STAMP, less specific to OCs, was higher in unstimulated H9 compared to RAW264.7 (~10 times) (Figure 3B). However, unstimulated J8 had an even higher expression of DC-STAMP (~17 times). After RANKL stimulation, the expression levels of DC-STAMP were similar in all three clones. The fusion/acidification marker ATP6v0d2 exhibited similar mRNA expression levels in all unstimulated clones and increased mRNA expression in RANKL-stimulated H9 and RAW264.7 compared with J8 (Figure 3B). Lastly, the fusion/podosome marker CD44, which has also been implicated in migration, cell–cell interactions and OC-physiology [25,26], displayed a slightly different pattern. RANKL stimulation did cause a subtle increase in CD44 expression in RAW264.7 but not in H9 or J8. However, CD44 expression was higher in both unstimulated and RANKL-stimulated H9 (~1500 times) and J8 (~1800 times) compared with RAW264.7 (Figure 3D). This is in line with previous studies suggesting that CD44 is not regulated by RANKL [27].

Podosome/adhesion/sealing zone marker β3 integrin [26] were more highly expressed in both unstimulated H9 (~1.6 times) and J8 (~20 times) compared with parental RAW264.7. After RANKL-stimulation, however, expression in parental RAW264.7 increased over 2000 times and RANKL-stimulated H9s had the same expression levels of β3, while J8 displayed slightly lower expression (Figure 3E).

The OC acidification marker ATP6i [28] had an expression pattern similar to OC-STAMP, with higher expression in unstimulated H9 (~2 times) compared to RAW264.7. Nevertheless, after RANKL stimulation expression levels in H9 and parental RAW264.7 were equal (Figure 3F) while J8 had significantly lower expression of ATP6i (~2.6 times).

### 2.4. H9 Forms TRAP-Positive OCs Cells Faster than Parental RAW264.7 Due to Faster Increase in TRAP-Gene Expression

The expression of OC-genes could not describe in sufficient detail the differences we observed in resorption, so the differentiation of the sub-clones was examined by a time course spanning the differentiation process. As unstimulated H9 have higher expression levels of several OC markers and after RANKL stimulation formed resorption pits earlier than RAW264.7 (Appendix A), our hypothesis was that H9 forms OCs faster upon RANKL stimulation than RAW264.7.

In response to RANKL stimulation, H9 started forming TRAP-positive multinuclear cells at Day 2 (Figure 4A,B) while RAW264.7 started to form TRAP-positive multinuclear osteoclast-like cells between Days 3 and 4 (Figure 4A,B). However, at Day 4 the number of TRAP-positive OCs was the same in RANKL-stimulated H9 and RAW264.7 (i.e., there was a catch-up effect in RAW264.7 between Days 3 and 4). TRAP gene expression analysis confirmed that H9 cells express higher amounts of TRAP mRNA after one-day of RANKL stimulation compared with parental RAW264.7, but the difference was insignificant at Day 2–4 (Figure 4C), again indicating that there is a catch-up effect in RANKL-stimulated RAW264.7.

H9 appeared to contain some small TRAP-positive cells already before RANKL stimulation (arrow Figure 4A day 1 H9). The number of presumably mononuclear TRAP-positive cells seemed to increase in unstimulated H9 over a 4-day time course and increasing cell density (arrows Figure 4A day 4 H9). The parental RAW264.7s also formed multinuclear cells but these were not TRAP-positive, indicating a different identity from OCs.

Furthermore, RANKL-stimulated J8 did not form either TRAP-positive mono- or multinucleated cells, but number of cells seemed higher than in the other cultures (Figure 4A). TRAP mRNA expression was slightly increased upon RANKL treatment, but about 20-fold less than parental RAW264.7 and H9 (Figure 4C).

### 2.5. Unstimulated H9 Displays Higher Gene Expression of OC Transcription Factor NFATc1 Compared to RAW264.7

To investigate why H9 forms OCs faster than the RAW264.7 gene, the mRNA expression of the genes involved at different stages of osteoclastogenesis were measured. Survival and proliferation cytokine M-CSF expression was higher in J8, compared to RAW264.7 in both un- and RANKL-stimulated cells (Figure 5A). On the other hand, M-CSF receptor c-fms expression was lower in J8 compared to RAW264.7 in both non-stimulated and RANKL-stimulated cells (Figure 5B), likely due to M-CSF induced c-fms endocytosis and degradation common to macrophages [29].

In both H9 and J8, the gene expression of RANKL receptor, RANK, was lower than in RAW264.7 in unstimulated cells (Figure 5C). After RANKL stimulation, RANK expression was increased in both H9 and J8, resulting in equal expression compared to RAW264.7. The gene expression of the main transcriptional regulator of osteoclastogenesis, transcription factor NFATc1 [17], was approximately three times higher in unstimulated H9 than RAW264.7 (Figure 5D). Unstimulated J8 had lower expression of NFATc1 (~0.4 times) compared to RAW264.7. After RANKL stimulation, this difference was abolished, and H9 and RAW264.7 had the same mRNA expression levels of NFATc1.

### 2.6. H9 Has Higher Degree of Nuclear Translocation of OC Transcription Factor NFATc1 than RAW264.7

Since H9 had a higher NFATc1 mRNA expression than RAW264.7, it was investigated whether NFATc1 protein was active (i.e., has it undergone nuclear translocation). In the nucleus, NFATc1 initiates osteoclastogenesis by upregulating first its own expression and thereafter increasing the expression of many OC markers [16]. The translocation of NFATc1 was more pronounced in the H9 sub-clone compared with RAW264.7 in both unstimulated and RANKL-stimulated cultures (Figure 6).

NFATc1 protein was detected in all unstimulated cells (i.e., H9, RAW264.7 and J8; Figure 6A), although in J8 the staining was very weak. In H9, both in multinuclear and mononuclear cells NFATc1 was localized to both the nucleus (active NFATc1) and cytoplasm (inactive NFATc1; arrows Figure 6A,B). Conversely, in unstimulated RAW264.7 and J8, NFATc1 was mainly confined to the cytoplasm (inactive NFATc1; Figure 6B). In RANKL-stimulated RAW264.7 and H9-derived OCs, NFATc1 translocated to the nucleus and there was a tendency that this nuclear translocation might be higher in H9 compared with RAW264.7. Furthermore, in RANKL-stimulated RAW264.7 cultures, there were many multinuclear cells without nuclear translocation of NFATc1. Also, RANKL-stimulated mononuclear J8 showed an increased nuclear translocation of NFATc1 compared to unstimulated J8 (Figure 6A–C); however, it was significantly lower than in RANKL-stimulated H9 cells. However, mononuclear J8 cells were positive for NFATc1 in the cytoplasmic compartment (Figure 6B).

## 3. Discussion and Conclusions

This study aimed to explore the heterogeneity of the mouse macrophage RAW264.7 cell line and if an OC-precursor could be isolated and enriched using initial gene expression levels of unstimulated sub-clone. The aim of the study was to establish a simple isolation protocol that could improve the performance of RAW264.7 cell line as a model of OCs.

Previously, it has been shown that it is possible to isolate OC-precursor sub-clones from RAW264.7 with different gene expression of the OC markers CtsK and TRAP in unstimulated and RANKL-stimulated sub-clones [22]. In the current study, it was possible to rank unstimulated RAW264.7 sub-clones by their TRAP and CtsK mRNA expression. In line with a previous study [22], there was a higher number of unstimulated sub-clones with high CtsK mRNA expression compared to high TRAP mRNA expression. Least frequent were sub-clones with both low CtsK and TRAP mRNA expression (i.e., only one clone with this phenotype was isolated). This indicates that the majority of RAW264.7 sub-clones likely exhibit some but variable potential for OC-like differentiation. Conversely, RANKL-stimulation was very inefficient in inducing the formation of multinucleated TRAP-positive cells in J8, the only sub-clone with low CtsK and TRAP mRNA expression. There are also sub-clones present in the RAW264.7 cell line lacking the ability to differentiate into OC-like cells after RANKL stimulation. The scarcity of low CtsK and TRAP expression in comparison to the parental line was unexpected but could reflect the selection bias for proliferating cells in the parental line, as OC differentiation and proliferation are mutually exclusive properties. This suggests that sub-cloning could indeed produce a more homogenous OC precursor population by eliminating sub-clones not able to differentiate into OCs upon RANKL stimulation. The apparent difference in TRAP-staining of the unstimulated H9 may be due to limited detection range of the assay or temporal changes in processing of TRAP 5a to the more active TRAP 5b and secretion of TRAP isoforms. Passaging the cells will eventually change the phenotype of macrophage-like cells, and therefore, we used low (<10) passage number in this study. Moreover, our method of sub-cloning followed by a screen of TRAP and CtsK expression can be used to revitalize and enhance OC-differentiation of cell lines capable of forming OCs.

Any cell line model of a mature OC needs the capability to carry out the specialized functions characterizing OCs, specifically the resorption of bone, consisting of demineralization of bone hydroxyapatite and degradation of collagen. RANKL-stimulated RAW264.7 have been shown to acidify and dissolve hydroxyapatite [22] and to form resorption pits [18,30,31]. Nevertheless, it has been debated whether RAW264.7 actually resorbs bone. RAW264.7 and sub-clone H9 were able to acidify to the same degree; however, H9 cultures seemed to form more sealing zones. Since demineralization likely requires the formation of polarized OCs with ruffled borders [32,33], formation of sealing zones and demineralization suggests that H9 and RAW264.7 can form polarized OCs.

In addition to demineralization, sub-clone H9 and RAW264.7 also form resorption pits, thus further suggesting that under appropriate conditions H9 and RAW264.7 do resorb bone. Interestingly, H9 formed deeper resorption pits faster than RAW264.7, which could be an advantage in an experimental setup. Moreover, it has been shown that in the absence of CtsK, OCs form resorption pits that are more shallow but larger in area, indicating deficient collagen degradation [34]. Furthermore, upon RANKL stimulation, both H9 and RAW264.7 produce detectable levels of the collagen degradation product CTX-I fragments. Therefore, the combined findings on the similarities and differences of H9s and RAW264.7s indicate that both degraded the mineral and the organic components of bone, and also that the measured difference in resorption could be due to lower CtsK expression in the parental RAW264.7.

In line with the higher mRNA expression of CtsK and TRAP in unstimulated H9 compared with RAW264.7, the mRNA levels of several other OCs markers were elevated already in unstimulated H9. As expected, higher expression for OC-specific genes (e.g., OC-STAMP, ATP6i and ATP6v0d2) were measured in H9, while less specific OC-related genes (e.g., DC-STAMP and CD44) were higher in J8. However, after 4 days of RANKL stimulation mRNA expression of the OC-markers in H9 and RAW264.7 were similar. In summary, in experiments without RANKL-stimulation, the expression of OC-specific genes was significantly higher in sub-clone H9 and, typically, expression was also highest in RANKL-stimulated H9. However, the proportional difference between stimulated and unstimulated was highest in parental RAW264.7 cells, implying that they undergo the largest shift in expression of OC-specific genes upon RANKL-stimulation. This indicates an initial lower degree of differentiation and a higher potential to differentiate in any direction, but also, differentiation to OCs would occur slower when compared with the H9.

One reason behind this pattern was revealed by time course of differentiation following RANKL stimulation—that is, H9s are faster in forming OCs. The H9 cells form multinucleated TRAP-positive OCs already at Day 2 compared to Days 3–4 for RAW264.7. In more detail, the sub-clone H9 has an initial phase of OC differentiation between Days 1–2 which is lacking in RAW264.7. This is then followed by a second phase of OC differentiation between Days 3–4, which also occurs in RAW264.7. From these data, we hypothesize that H9 is more committed to the OC-lineage than the parental RAW264.7 population, but the parental population has the potential to differentiate into similar cells in most aspects. However, the large proportion of multinuclear TRAP-negative cells in unstimulated RAW264.7-cultures indicate that it may differentiate also to other directions in the absence of further stimulus.

Considering OC-differentiation, mRNA expression of key osteoclastogenesis genes revealed that the most likely determinant of the difference between H9 and RAW264.7 was an elevated gene expression of the osteoclastogenesis master transcription factor, NFATc1, in unstimulated H9. Furthermore, unstimulated H9 also exhibits higher nuclear translocation of NFATc1, indicating that there is more active NFATc1 in unstimulated H9 compared to RAW264.7. On the contrary, gene expression of RANK was actually lower in H9 compared with RAW264.7, supporting the idea that RANK mRNA expression is not always correlated to the effect of RANKL stimulation in RAW264.7-derived cells [22]. Combined, these data strongly suggest that H9 is a more committed OC precursor compared to the parental RAW264.7.

Sub-clone J8, selected for its low mRNA expression of TRAP and CtsK, did not exhibit any OC-like features. Instead, J8 presented a more macrophage-like phenotype, with low levels of TRAP+ cells, multinucleation, demineralization, resorption pit formation and expression of OC markers, but with high capacity for phagocytosis, also after RANKL stimulation. An inability of certain sub-clones of RAW264.7 to form multinuclear TRAP-positive OC-like cells has been reported before [22] indicating the extent of heterogeneity in RAW264.7. Although J8 did not form multinucleated TRAP-positive cells or exhibit other OC-characteristics, it still responded to RANKL stimulation, since mRNA expression of M-CSF and its receptor were inversely regulated upon RANKL stimulation, suggesting a more active receptor signaling, consistent with a macrophage phenotype. Furthermore, an increased expression of OC-specific genes was also seen in J8. Still, nuclear translocation of NFATc1 seemed to be very low in J8. Characterization of J8 shows that J8 exhibits a macrophage phenotype dissimilar from the parental RAW264.7, one that is unable to fully differentiate to an OC. However, macrophage phenotypes are diverse and plastic (e.g., in vivo also TRAP-positive macrophages are common [35,36,37]). Therefore, J8 does not represent macrophages in general.

In conclusion, RAW264.7 sub-clone H9 represent a more homogenous and OC-committed cell than the parental RAW264.7 cell line with OC features such as multinucleation, TRAP expression, demineralization and collagen degradation. Furthermore, H9 forms OCs in a short time span (i.e., on plastic TRAP-positive multinuclear cells appear in 2 days compared with 4 days for RAW264.7), and the first indications of resorption are observed after 3 days. This enables differentiation of a large number of OCs within a couple of days, thus reducing experiment time and reagent amounts. This could be especially valuable in initial larger screening experiments but also later in the scientific process to study osteoclastogenesis. Furthermore, we demonstrated that the expression of OC-genes, TRAP and CtsK, in unstimulated precursors can be used to purify heterogeneous OC-precursor populations and screen for clones that efficiently differentiate to OCs.

## 4. Material and Methods

### 4.1. Cell Culture

RAW264.7 and the selected clones were cultured in minimum essential medium (MEM-α) supplemented with 10% of heat inactivated fetal bovine serum (FBS), 100 µg/mL Gentamicin and 2 mM L-glutamine (all from Gibco, Life Technologies, Carlsbad, CA) referred to as cell culture media. All cells were incubated at 37 °C with 5% carbon dioxide (CO_2_).

### 4.2. RAW264.7 Single Cell Cloning

Single cell cloning of RAW264.7 was performed in 96-well plates (Nucleon, Delta surface, Nunc, Roskilde, Denmark). Single cell dilution was performed as follows (see also Appendix A); 1 × 10^6^ RAW264.7 cells were seeded into A1 and then serially diluted 1:2 in B1 to H1. This was followed by a second serial dilution in each row (e.g., A1–A12). Formation of single cell colonies was tracked, and the colonies later suspended by vigorous pipetting and expanded in T25 culture flasks (Sarstedt Inc, Leicester, UK), before freezing in cell culture medium supplemented with 10% dimethyl sulfoxide (DMSO) (Sigma, St. Louis, MO, USA). After selection, the sub-clones were expanded, and in the experiments the presented passage numbers were <10.

### 4.3. RANKL Stimulation

For gene expression analysis, RAW264.7 and sub-clones H9.2d5 (alias H9) and J8.2g6 (alias J8) were seeded at 5000 cells/well in 6-well plates (Nunc) +/− 10 ng/mL recombinant mouse RANK Ligand (R&D systems, Minneapolis, MN, United States) in cell culture media and cultured for 5 days. The medium was changed on every third day of culture.

### 4.4. RNA Purification and Reverse Transcription

For high-throughput screening and heat-mapping, the total RNA was purified using QIA Shredder + RNeasy Plus Mini or Micro Kit (Qiagen, Hilden, Germany) according to the manufacturer’s protocols. Quantification and the purity of the total RNA was determined using a NanoDrop Spectrophotometer ND-1000 (Thermo Scientific, Wilmington, DE, United States). Reverse transcription was performed on the total RNA using the SuperScript III Reverse Transcriptase Kit (Life Technologies) or iScript cDNA synthesis kit (BioRad, Hercules, CA, United States) according to manufacturer’s protocols on a Gene Amp^®^ PCR System 9700 (Life Technologies, Carlsbad, CA, United States). 

### 4.5. Real Time qPCR 

PCR was run in duplicates of 10 µL with 1*iTaq™ Universal SYBR^®^ Green Supermix (BioRad, Hercules, CA, USA) or 1X KAPA SYBR FAST qPCR Master Mix (KAPA BioSystems, Wilmington, MA), 900 nM primer (Appendix A) in Hard-Shell^®^ High-Profile 96-Well Semi-Skirted PCR Plates (BioRad) sealed with Microseal ‘B’ Adhesive Seal (BioRad). The samples were run on the CFX96 Real Time PCR Detection System (Bio-Rad) according to the following process: 95 °C for 3 min followed by 40 cycles of 95 °C for 5 s, 60 °C for 5 s and 72 °C for 10 s followed by melt curve analysis. Analyses were made using CFX Manager 3.0 (Bio-Rad) and heat mapping was done using MultiExperiment Viewer 4.9 [38]. 

### 4.6. TRAP Staining of OC Cultures

Samples were fixed with 4% formalin at room temperature for 30 s and then stained for tartrate resistant aid phosphatase activity using the Leukocyte acid phosphatase (TRAP) kit (Sigma-Aldrich Merck, Darmstadt, Germany) according to manufacturer’s protocols. 

The cells were imaged using a Nikon Eclipse TE300 microscope (10X objective, Nikon, Stockholm, Sweden) and analysed using NIS-Elements software (Nikon, AR 4.30.01, Tokyo, Japan). TRAP-positive cells larger than 300 μm^2^ (i.e., osteoclasts), were quantified using the ImageJ [39] macro that counted cells gated positive for high staining intensity for red and blue RGB channels and size of stained area (code in Appendix A).

### 4.7. Demineralization Assay

1 × 10^4^ cells/well of parental RAW264.7 and sub-clones H9 and J8 were plated on Corning Osteo Assay plates (Sigma, St- Louis, MO, United States) in cell culture media and treated with 10 ng/mL of RANKL (R&D Systems) for 7–8 days. The medium was changed every third day of culture. Finally, the cells were removed using water on Day 8 and the area acidified was measured using the Nikon Eclipse TE300 microscope (10X objective; Nikon, Tokyo, Japan) and analysed using NIS-Elements software (Nikon, AR 4.30.01 Tokyo, Japan).

### 4.8. Phagocytosis Assay

RAW264.7, H9 and J8 cells were seeded in a cell culture medium on 96-well plates (nucleon Δ surface, NUNC, Copenhagen, Denmark) at 5 × 10^3^ cells/well. The cells were treated with 10 ng/mL RANKL and the cell culture medium was changed on the third day. On Day 5, the cells were washed 3 times with PBS and incubated for 2 h with 100 µg/mL of fluorescent Alexa-488-labelled *Escherichia coli* bioparticles (Invitrogen Co., Carlsbad, CA, United States) in the cell culture medium. The cells were washed 3 times with PBS and the internalized bacteria was measured as an average value of 4 × 4 matrices and read at 490 nm using ClarioStar (BMG Labtch GMBH, Ortenburg, Germany) at room temperature.

### 4.9. Immunocytochemistry

RAW264.7, H9 and J8 (5 × 10^3^ cells/well) were grown in Lab-Tek II 8-well (Sigma-Aldrich, Saint Louis, MI, United States) plates in a cell culture medium for 4 days in 10 ng/mL mouse recombinant RANK Ligand (R&D systems). The cells were then washed twice with PBS, fixed with 4% paraformaldehyde for 15 min at 37 °C, and washed with PBS. After permeabilization with 0.1% Triton X-100 in PBS for 15 min, 0.1% BSA in PBS was added to the cells for another 60 min, both at room temperature. The cells were then stained overnight at 4 °C with anti-NFAT2 antibody (1:300) (#: ab2796, Abcam, Cambridge, UK). The wells were washed in PBS for 10 min 3 times, and the cells were stained with goat anti-mouse Alexa-488 (1:100) along with Hoechst 33342 (1:7000) and phalloidin-568 (1:400) (all Thermo Fisher Scientific, Waltham, MA, United States) overnight at 4 °C. The samples were washed 3 times (10 min/wash) in PBS and mounted onto glass microscope slides with ProLong Diamond (Thermo Fisher Scientific, Waltham, MA, United States) mounting medium. Z-stacks were captured with Nikon A1R+ confocal laser microscope system (NIKON) using 20X objective, a pixel size fulfilling Nyquist sampling theorem and NIS-Elements (Nikon, AR 4.30.01 Tokyo, Japan). Laser power and detector gain were adjusted to cover the widest possible range of intensity values for calculation of Pearson’s correlation coefficient (PCC). Three-dimensional colocalization was measured from the entire captured z-stacks as PCC with the ImageJ colocalization plugin using automated Costes approximation of background (Wright Cell Imaging Facility, UHN, Toronto, ON, Canada).

### 4.10. Bone Chips

Rods (6 mm in diameter) of bovine femoral cortical bone were cut with the ISOMET Low Speed Saw (Buehler, Esslingen, Germany) into 100–150 µm thick slices. The slices were washed by ultra-sonication for 20 min in 70% ethanol and rinsed extensively in distilled water. For long-term storage, the slices were kept in 20% ethanol at 4 °C.

### 4.11. Bisphosphonate Staining and Resorption Pit Imaging

The cells were plated on the bone discs at a density of 2 × 10^4^ cells/well and treated with +/− 10 ng/mL mouse recombinant RANK Ligand (R&D Systems) for different time points. After the third day of culturing, the medium was acidified. The medium was changed every third day, coinciding with the collection and fixation with 4% formalin for 30 s. The fixed bone chip cultures were permeabilized with 0.1% Triton X/PBS for 15 min and incubated with fluorescent bisphosphonate (1:1000, kindly provided by Dr. Fraser Coxon, University of Aberdeen, Aberdeen, UK), Alexa Fluor 568 phalloidin (1:400; Thermo Fisher Scientific, Waltham, MA, United States) and Hoechst 33342 (1:3000; Thermo Fisher Scientific, Waltham, MA, United States) in 0.1% BSA/PBS for 30 min. The discs were washed twice in PBS, once in distilled water and mounted onto a microscope slide using the Prolong Diamond hard set mounting medium (Thermo Fisher Scientific, Waltham, MA, United States) and left to dry overnight at room temperature in the dark. Z stack images were acquired on the Nikon A1R+ confocal laser microscope system (Nikon, Tokyo, Japan) using 20X objective and NIS-Elements (Nikon, AR 4.30.01, Tokyo, Japan). The resorbed area was measured along the pit edges without considering the pit depth. Three fields of view from technical duplicates from 3 independent experiments were quantified. The pit depth was measured from the z-stacks by selecting the section with the highest fluorescence intensity at the pit bottom and measuring the difference between that and the surface of the un-resorbed bone using the section thickness. From the technical replicates from 3 independent experiments, the 8 deepest pits for each field of view were quantified—a total of 72 resorption pits per sub-clone for each time point. The resorbed volume was calculated by assuming a semi-ellipsoidal geometry of the resorption pit (Formula (1)).
(1)$$V_{RP}=\frac{2}{3}\times h_{depth}\times A_{RP}$$

### 4.12. Measurement of Type I Collagen Degradation Marker (CTX-I)

The cells were seeded at 2 × 10^5^ cells/well on bone chips in a cell culture medium. After 3 days of culturing, the cell culture medium was changed to an acidified medium (cell culture medium with a final pH of 6.5 (= 0.085% HCl)) and the cells were treated with +/− 10 ng/mL of recombinant mouse RANKL (R&D systems). The medium was collected on Day 3, 6, 9 and 12 after the start of culture. The bone resorption activity was determined by quantifying the C-terminal telopeptide degradation product of type I collagen in 50 L aliquots of the culture supernatants using CrossLaps^®^ for culture (CTX-I) ELISA (IDS, Tyne and Wear, UK) according to the manufacturer’s instructions. The CTX-I concentrations present in the medium from cells on the plastic and bone slices without cells were subtracted.

### 4.13. Transmission Electron Microscopy (TEM)

RAW264.7 cells were seeded at 2 × 10^4^ cells/well on dentine and treated with 10 ng/mL mouse recombinant RANK Ligand (R&D systems). The RAW264.7 cells on dentine were fixed with 2.5% glutaraldehyde in 0.1 M cacodylate buffer pH 7.4 overnight at 4 °C. The discs were then decalcified in 2% glutaraldehyde in 0.1 M cacodylate buffer with 0.1 M EDTA pH 7.4, with changes every 2–3 days for approximately 3 weeks. Following decalcification, the discs were cut into quarters and processed for routine TEM as follows: the discs were first placed in 1% osmium tetroxide in water for 1 h, followed by 1 h in 1% uranyl acetate in H_2_O. The discs were then dehydrated through a graded series of ethanol and embedded in epoxy resin, which was polymerized at 60 °C for 2 days.

Once polymerized, the resin blocks containing the dentine discs were trimmed for ultramicrotomy using Leica EM UC7 (Leica Microsystems, Wetzlar, Germany). Initially, 500 nm-thick survey sections were stained with toluidine blue and examined by light microscopy to confirm the presence of resorbing osteoclasts on the dentine. Once an osteoclast was located, 200–300 nm-thick sections were cut and collected onto formvar coated copper grids. The sections were counterstained in Leica EM AC20 (Leica Microsystems, Wetzlar, Germany) with standard solutions of uranyl acetate and lead citrate to enhance contrast. Electron micrographs were taken using the JEOL 1400 Plus transmission electron microscope equipped with UltraVUE camera (AMT, Woburn, MA, USA).

## Figures and Tables

**Figure 1 ijms-21-00538-f001:**
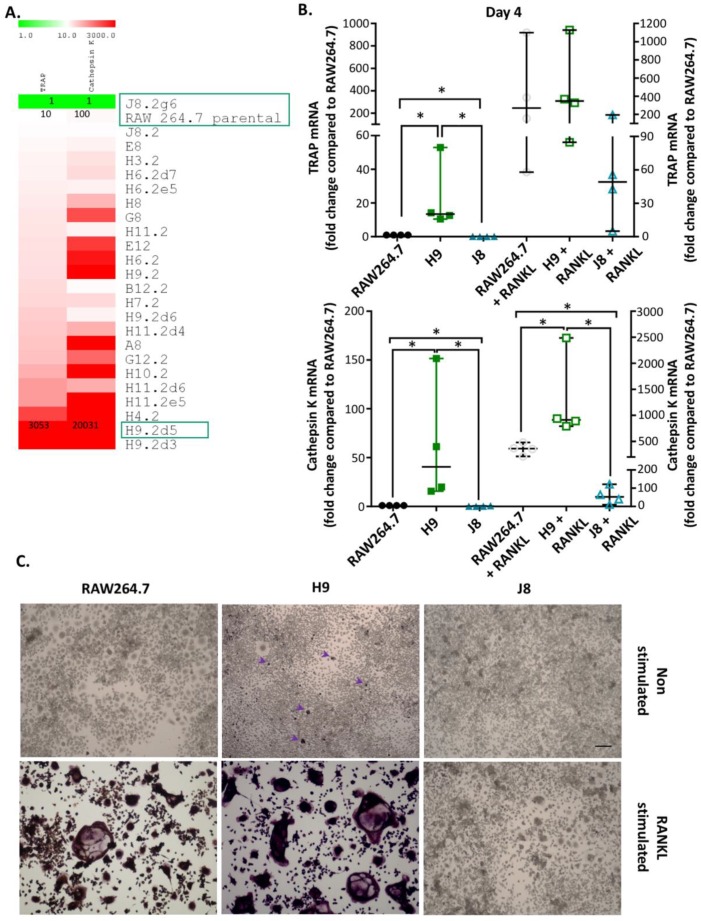
Single cell cloning and osteoclast gene expression screening for the selection of sub-clones. (**A**) Representative heat map of TRAP and CtsK gene expression in RAW264.7 clones using qPCR. (**B**) TRAP and CtsK gene expression in RAW264.7 and sub-clone H9 and J8 +/− 10 ng/mL RANKL for 4 days (*n* = 4). Data were analyzed using Mann–Whitney U test. * *p* < 0.05. Values are given as median +/− range. (**C**) TRAP activity staining of RAW264.7 and sub-clone H9 and J8 +/− 10 ng/mL RANKL for 4 days. Arrows = mononuclear TRAP+ cells. Arrows point to multinuclear TRAP+ cells. Scale bar is 200 µm and all micrographs have the same magnification.

**Figure 2 ijms-21-00538-f002:**
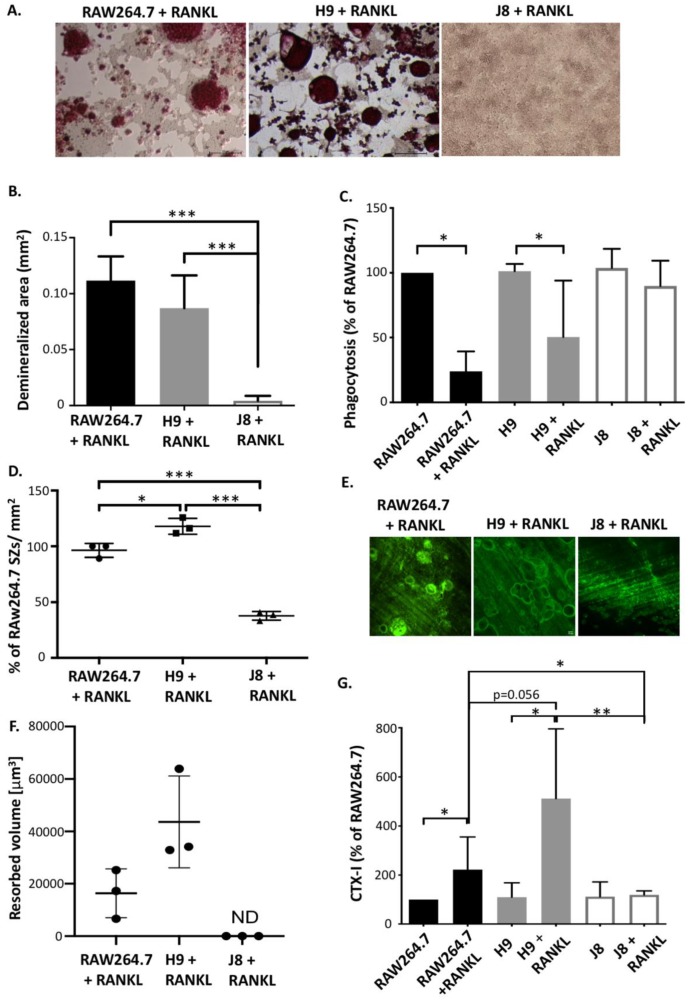
Osteoclast characteristics in RAW264.7 and sub-clones H9 and J8 in response to RANKL stimulation. (**A**) TRAP activity staining on the surface of hydroxyapatite wells showing area that had been acidified after 8 days. Scale bar 100 μm, all micrographs have the same magnification. (**B**) Quantification of the acidified area after 8 days (mean ± SD, *n* = 3). (**C**) Quantification of phagocytosis capacity after 5 days of differentiation with M-CSF or RANKL (mean ± SD, *n* = 4). Data were analyzed using Mann–Whitney U test. * = *p* value 0.05. (**D**) Quantification of sealing zones stained for f-actin on bone-coated coverslip on day 10. Quantification was done using ImageJ (mean ± SD, *n* = 3). (**E**) Quantification of the resorption pit volumes. *n* = 3. Scale bar 50 μm, all micrographs have the same magnification. (**F**) Bisphosphonate staining to identify resorption pits after 6 days of culture. (**G**) CTX-I measurement on Day 6. Statistical analysis with one-way ANOVA with Tukey’s multiple comparison * *p* < 0.05, ** *p* < 0.01, *** *p* < 0.001.

**Figure 3 ijms-21-00538-f003:**
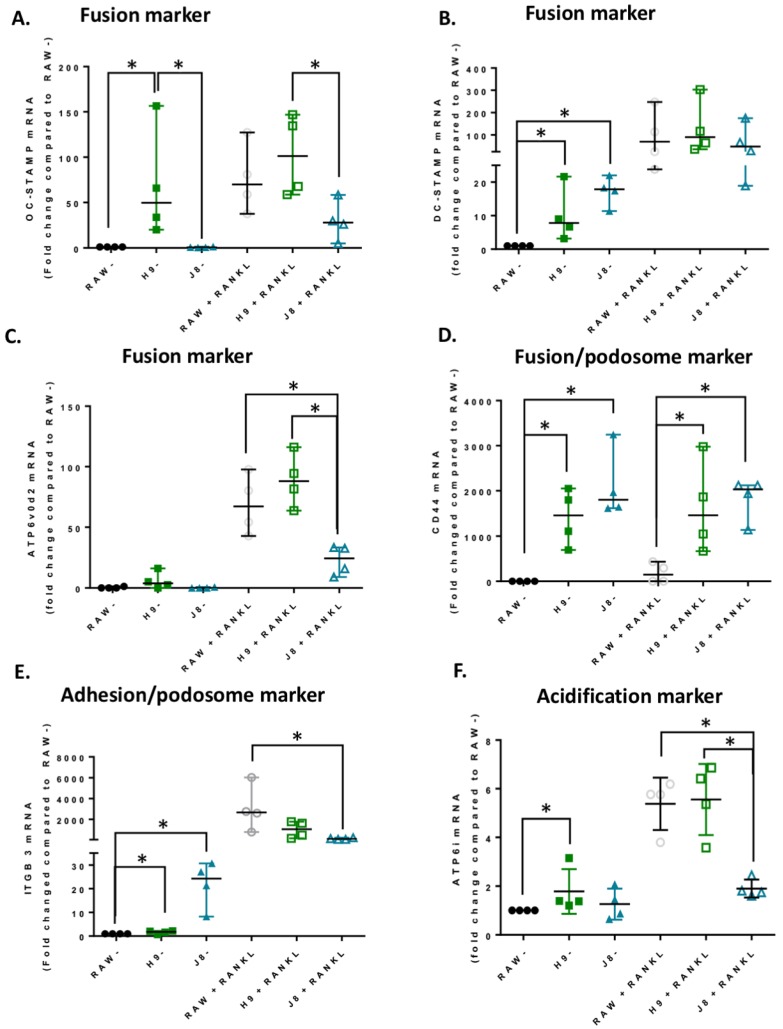
Osteoclast gene expression in RAW264.7 and sub-clones H9 and J8. RAW264.7, H9 and J8 were stimulated +/− 10 ng/mL RANKL for 4 days after which mRNA expression of osteoclast (OC) genes was measured (*n* = 4). (**A**) OC-stamp mRNA, (**B**) ATP6v0d2 mRNA, (**C**) DC-stamp mRNA, (**D**) CD44 mRNA, (**E**) integrin b3 mRNA, (**F**) ATP6i mRNA, Data were analyzed using Mann-Whitney U test. * *p* < 0.05. Values are given as median +/− range.

**Figure 4 ijms-21-00538-f004:**
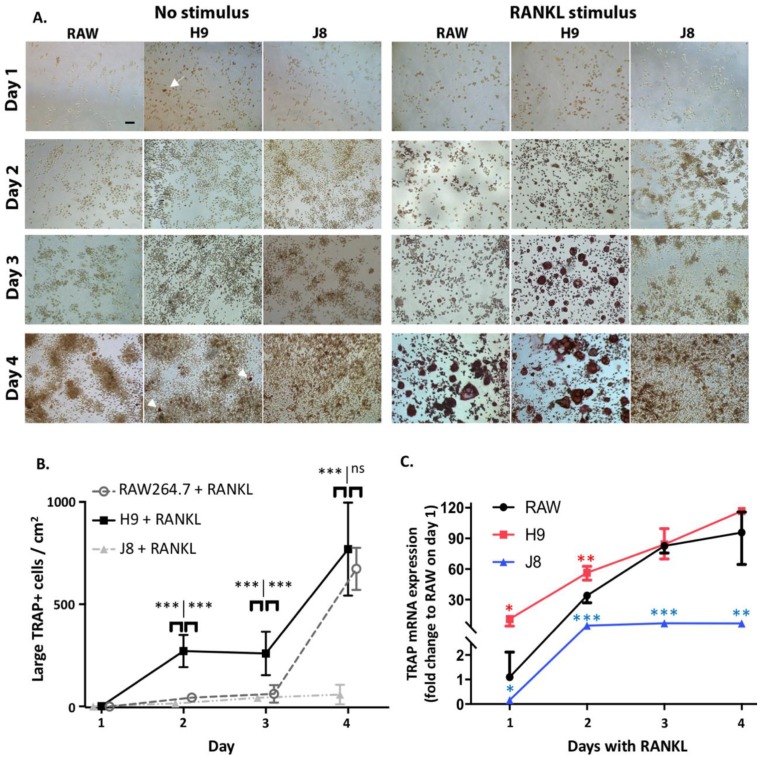
Time-dependent formation of osteoclasts in unstimulated and RANKL-stimulated RAW264.7 and sub-clones H9 and J8. (**A**) TRAP staining of unstimulated and RANKL-stimulated RAW 264, H9 and J8 cells after 1–4 days. Scale bar 100 μm, all subplots have the same magnification. Arrows point to TRAP-positive cells on Day 1 and TRAP-positive multinuclear cells on Day 4. (**B**) Quantification of the number of large TRAP+ cells/cm^2^ in RANKL-stimulated RAW264.7, H9 and J8 cells (mean ± SD, *n* = 3). The 100-pixel area corresponds to an area of 85 µm^2^ and the average size of a mononuclear cell is between 50–300 µm^2^, while the threshold for large (multinuclear) cells was set for 300 µm^2^. Statistics were done by one-way ANOVA followed by Turkey’s multiple comparison test (**C**) TRAP mRNA expression in RANKL-stimulated RAW264.7, H9 and J8 cells after 1–4 days (*n* = 3). Statistics were done with two-tailed multiple *t*-test comparing the sub-clones to parental line. * *p* < 0.05, ** *p* < 0.01, *** *p* < 0.001.

**Figure 5 ijms-21-00538-f005:**
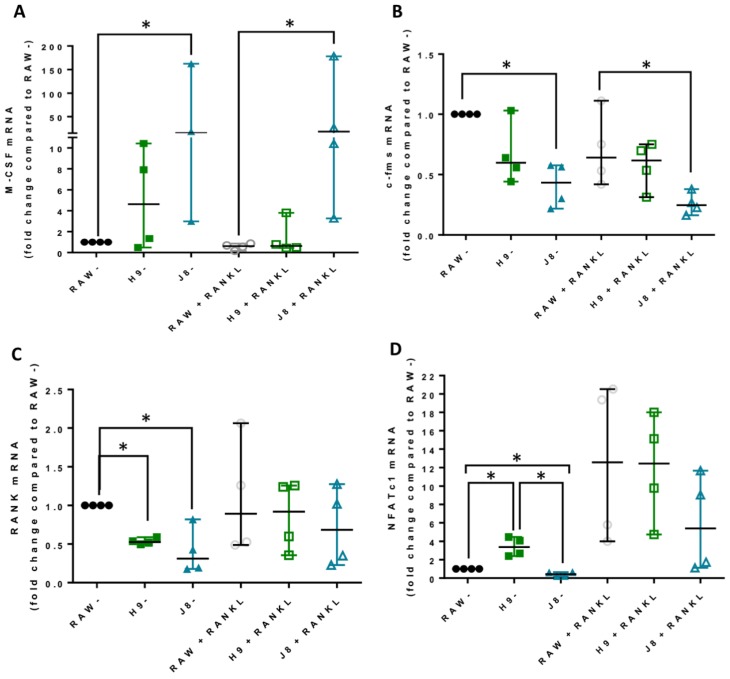
Gene expression of genes involved in osteoclastogenesis in unstimulated and RANKL-stimulated parental RAW264.7 and sub-clones H9 and J8. RAW264.7, H9 and J8 were stimulated +/− 10 ng/mL RANKL for 4 days after which mRNA expression of OC genes were measured (*n* = 4). (**A**) M-CSF mRNA, (**B**) c-fms mRNA, (**C**) RANK mRNA and (**D**) NFATc1 mRNA. Data were analyzed with Kruskal Wallis followed by Dunn’s multiple comparison test. * *p* = 0.05. Values are given as median +/− range.

**Figure 6 ijms-21-00538-f006:**
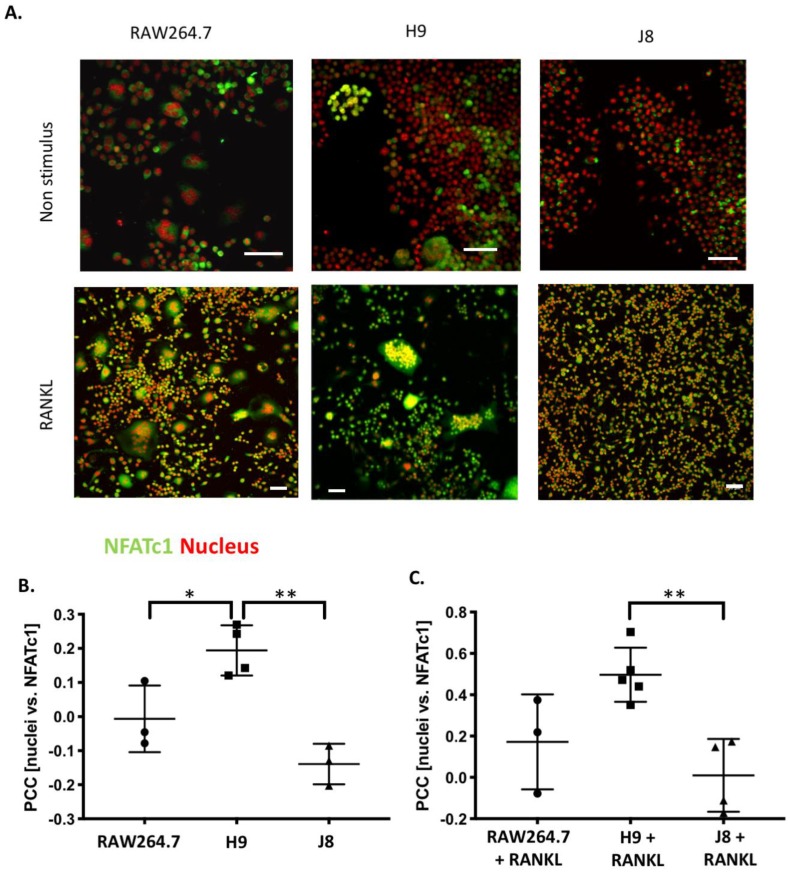
Nuclear translocation of NFATc1 in unstimulated and RANKL-stimulated RAW264.7, H9 and J8. (**A**) RAW264.7, H9 and J8 were stained for NFATc1 (Anti-NFAT2 antibody [7A6] (green) and nuclei with Hoechst 33342 (red)). Active NFATc1 translocated to the nucleus is represented by the yellow color. Scale bar 50 μm. (**B**) Pearson correlation of NFATc1 and cell nucleus in unstimulated cultures and (**C**) in cultures with 10 ng/mL RANKL. Quantification was done using ImageJ (mean ± SD, *n* = 3), statistical analysis with multiple comparison ANOVA followed by Turkey´s multiple comparison test * *p* < 0.05, ** *p* < 0.01, *n* = 3.

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
