# Peer review of "A Sub-Clone of RAW264.7-Cells Form Osteoclast-Like Cells Capable of Bone Resorption Faster than Parental RAW264.7 through Increased De Novo Expression and Nuclear Translocation of NFATc1"

_ijms, 2020, doi:10.3390/ijms21020538_

Round 1

Reviewer 1 Report

The paper is interesting, timely and useful, since it identifies a sub-clone of the RAW264.7 cell line that, under the proper conditions, rapidly differentiates to large numbers of osteoclasts. This is very important for all scientists working in the field of osteoclastogenesis since one of the hurdles is precisely the poor osteoclastic differentiation of this cell line due to its heterogeneity, as the authors correctly point out. Apart from a few typos, the paper is very well written. The paper is scientifically sound.

Author Response

The paper is interesting, timely and useful, since it identifies a sub-clone of the RAW264.7 cell line that, under the proper conditions, rapidly differentiates to large numbers of osteoclasts. This is very important for all scientists working in the field of osteoclastogenesis since one of the hurdles is precisely the poor osteoclastic differentiation of this cell line due to its heterogeneity, as the authors correctly point out. Apart from a few typos, the paper is very well written. The paper is scientifically sound.

We thank the reviewer for the time contributed to reading our manuscript and the kind assessment of our research.

Reviewer 2 Report

The study aims to understand the osteoclast activities of sub-clones in RAW264.7 heterogeneous cell line. The authors identified sub-clone H9 which presented higher expression of OC-markers and resorption activities. They suggested that sub-clone H9 could be valuable in screening experiments.

Comments:

There are only a few TRAP-positive H9 cells under non-stimulated condition (Figure 1C). Given that subclone is derived from single cell cloning, it is expected the cells are homogenous and expressing TRAP. Line343: why there is a population of TRAP-positive cells, considering they are derived from the same single cell? Figure 1C: Are the cells appeared mononuclear in sub-clone J8 under RANKL-stimulated? Are there multinuclear/fusion cells observed? As shown in figure, it seems that the cells are clustered. Figure 1 legend: there is no double arrow shown in figure. Figure 2: Sub-clone J8 did not form TRAP-positive OC in Figure 2A. Why there is sealing zone in J8 cells (data not shown)? I agree with authors that increased resorption activities in H9 OC is probably due to increased OC differentiation ability. To test this hypothesis, parental and H9 Raw cells could be differentiated into mature OC on collagen plate. Mature OCs (considering 4 days of differentiation in parental line, and 3 days of differentiation in H9, as shown in figure 4) will be dissociated and plated on HA-coated plate/bone discs for resorption assays. How many passages of sub-clone H9 were tested and retained the increased OC activities? This would be important considering this study aim to establish a cell line for screening and future application.

Author Response

Comments and Suggestions for Authors

The study aims to understand the osteoclast activities of sub-clones in RAW264.7 heterogeneous cell line. The authors identified sub-clone H9 which presented higher expression of OC-markers and resorption activities. They suggested that sub-clone H9 could be valuable in screening experiments.

We express our gratitude for the time contributed to the review and the constructive criticism that has helped us to improve our manuscript.

Comments:

Point 1. There are only a few TRAP-positive H9 cells under non-stimulated condition (Figure 1C). Given that subclone is derived from single cell cloning, it is expected the cells are homogenous and expressing TRAP. 

Answer 1. The sub-clone H9 was screened for its expression of TRAP mRNA, however, the intensity of the staining is a compound measure of multiple things in post-translational processing of TRAP and assay sensitivity. We added a passage to the discussion to clarify this question: “The apparent difference in TRAP-staining of the unstimulated H9s may be due to limited detection range of the assay or temporal changes in processing of TRAP 5a to the more active TRAP 5b and secretion of TRAP isoforms.”

Furthermore, RAW264.7 is a macrophage-like cell line and one of the best cell lines to study osteoclast differentiation as well as macrophage activation. Macrophage activation is sensitive to cell’s microenvironment and a transient process, which can assume a different direction multiple times during a lifetime of a cell. Hence, the model replicates biology very well, but the biology of macrophages is difficult to classify in contrast to other white blood cells. That is why the single cell cloning resulted to a clone with predisposition to differentiate to an osteoclast, implying higher probability within a certain timeframe.

Point 2. Line343: why there is a population of TRAP-positive cells, considering they are derived from the same single cell?

Answer 2. We revised the text and no longer use the word population in this context. The apparently stochastic emerging of TRAP-positive cells in unstimulated conditions is in relation to both time and their microenvironment. In unstimulated conditions the microenvironment cannot be controlled, and therefore, the observation of some TRAP-positive cells is due to inherent predisposition of the H9-clone and not due to environmental factors.

Point 3: Figure 1C: Are the cells appeared mononuclear in sub-clone J8 under RANKL-stimulated? Are there multinuclear/fusion cells observed? As shown in figure, it seems that the cells are clustered.

Answer 3. Indeed, the J8 formed fewer multinuclear cells and at many occasions there were only mononuclear cells clustered together. However, some multinuclear cells were observed both on plastic and on bone.

Point 4. Figure 1 legend: there is no double arrow shown in figure.

Answer 4. We have revised the arrows to be more visible and changed the wording.

Point 5. Figure 2: Sub-clone J8 did not form TRAP-positive OC in Figure 2A. Why there is sealing zone in J8 cells (data not shown)?

Answer 5. The sub-clone J8 formed multinuclear cells with low frequency and on bone these were able to generate sealing zones. We have added representative images to supplementary material.

Point 6. I agree with authors that increased resorption activities in H9 OC is probably due to increased OC differentiation ability. To test this hypothesis, parental and H9 Raw cells could be differentiated into mature OC on collagen plate. Mature OCs (considering 4 days of differentiation in parental line, and 3 days of differentiation in H9, as shown in figure 4) will be dissociated and plated on HA-coated plate/bone discs for resorption assays.

Answer 6. The suggested experiment would indeed give definite proof that the difference we observe between the sub-clones is on differentiation and not on resorption. However, even the data already presented gives evidence to the same interpretation. We show significantly more rapid differentiation while measures of resorption (in our case compound effect of differentiation and resorptive ability) no longer give a significant difference between the sub-clone H9 and parental RAW264.7.

Moreover, within the 10-day revision period it is not feasible to include new experiments.

Point 7. How many passages of sub-clone H9 were tested and retained the increased OC activities? This would be important considering this study aim to establish a cell line for screening and future application.

Answer 7. We tested the sub-clones in passages below 10. Hence, the sub-clones retained their phenotype during at least 10 passages. Using similar low passage numbers is a common strategy with macrophage-like cell lines. Furthermore, we present a simple screening method to revitalize cell lines to provide more efficient osteoclast differentiation.

We have revised our manuscript to make a note of the passage numbers in methods and discussion.

Round 2

Reviewer 2 Report

The comments have been addressed appropriately.